# CWCD: Category-Wise Contrastive Decoding for Structured Medical Report Generation

**Shantam Srivastava**                 SS693@BUFFALO.EDU
**Mahesh Bhosale**                 MBHOSALE@BUFFALO.EDU
**David Doermann**                 DOERMANN@BUFFALO.EDU
**Mingchen Gao**                 MGAO8@BUFFALO.EDU

*The Department of Computer Science and Engineering*
*University at Buffalo, The State University of New York, NY, USA*

**Editors:** Accepted for publication at MIDL 2026

## Abstract

Interpreting chest X-rays is inherently challenging due to the overlap between anatomical structures and the subtle presentation of many clinically significant pathologies, making accurate diagnosis time-consuming even for experienced radiologists. Recent radiology-focused foundation models, such as LLaVA-Rad and Maira-2, have positioned multi-modal large language models (MLLMs) at the forefront of automated radiology report generation (RRG). However, despite these advances, current foundation models generate reports in a single forward pass. This decoding strategy diminishes attention to visual tokens and increases reliance on language priors as generation proceeds, which in turn introduce spurious pathology co-occurrences in the generated reports. To mitigate these limitations, we propose **C**ategory-**W**ise **C**ontrastive **D**ecoding (**CWCD**), a novel and modular framework designed to enhance structured radiology report generation (SRRG). Our approach introduces category-specific parameterization and generates category-wise reports by contrasting normal X-rays with masked X-rays using category-specific visual prompts. Experimental results demonstrate that CWCD consistently outperforms baseline methods across both clinical efficacy and natural language generation metrics. An ablation study further elucidates the contribution of each architectural component to overall performance.

**Keywords:** Radiology Report Generation, Multimodal Large Language Models, Contrastive Decoding, Chest X-rays.

## 1. Introduction

Over the past two decades, the rapid advancement of Artificial Intelligence (AI) has significantly improved automated interpretation of medical images (Sabri et al., 2025; Khalifa and Albadawy, 2024), particularly chest X-rays, which remain one of the most frequently performed diagnostic procedures worldwide (Broder, 2011). Chest X-rays are highly valued due to their low cost, minimal radiation exposure, and ability to provide substantial clinical information. Despite these advantages, generating radiology reports remains a cognitively demanding and time-consuming task (Lee et al., 2013). Compounding this challenge, the growing demand for interpreting chest X-rays has outpaced the supply of radiologists (Christensen et al., 2025), leaving many radiologists overworked and vulnerable to fatigue (Vosshenrich et al., 2021).

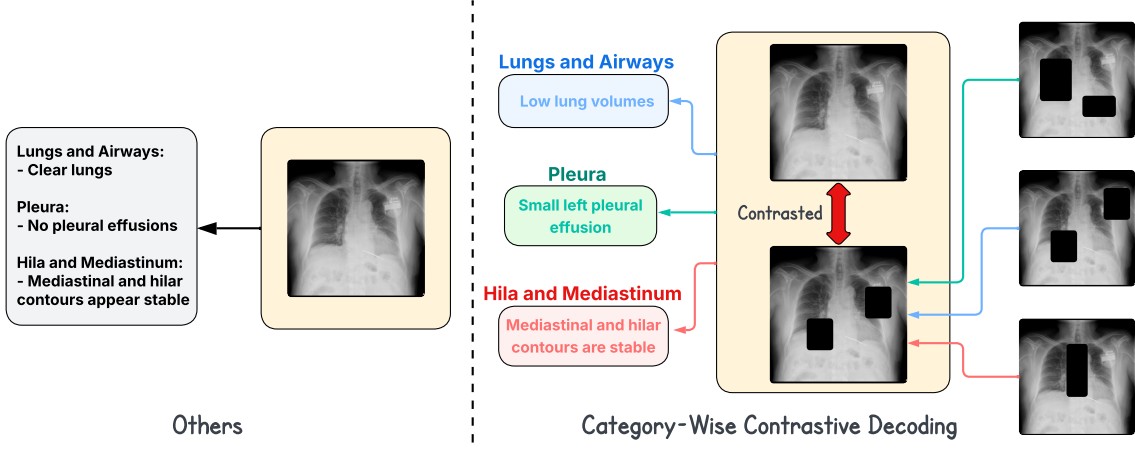

Figure 1: Category-Wise Contrastive Decoding (CWCD) generates a category-wise structured report under eight anatomical headers by contrasting a normal X-ray with a masked X-ray (3 categories shown here for brevity).

Automated Radiology Report Generation (RRG), the task of producing free-text descriptions of visual observations from a radiology image, such as a chest X-ray, has therefore emerged as an essential research direction (Jing et al., 2018b; Li et al., 2025). However, automated RRG remains fundamentally challenging: unlike natural images, chest X-rays exhibit low contrast and may contain subtle, highly localized pathologies. The requirement to generate long, unconstrained textual reports imposes additional demands on model fidelity. Unlike visual question answering, which operates within relatively short, focused outputs, comprehensive radiology findings reports may exceed 200 tokens and the model must reason jointly over multiple, often overlapping, anatomical regions.

Early encoder-decoder approaches (Yuan et al., 2019; Jing et al., 2018a) established a strong foundation and were able to generate linguistically cohesive reports, however, they often lagged in clinical efficacy (Yang et al., 2022). The rise of Large Language Models (LLMs) (Radford et al., 2019; Touvron et al., 2023) and subsequently multi-modal LLMs (MLLMs) (Liu et al., 2023; Alayrac et al., 2022) enabled the development of the first generation of radiology foundation models (Wu et al., 2023; Chen et al., 2024; Hyland et al., 2023; Pellegrini et al., 2025; Wang et al., 2023). These models leveraged the superior language modeling and linguistic reasoning capabilities of LLMs and substantially scaled parameter counts to surpass the then state-of-the-art encoder-decoder models. They delivered remarkable improvements in clinical efficacy metrics and demonstrated stronger generalization performance on out-of-distribution datasets (Pellegrini et al., 2025).

The second generation of radiology foundation models further advanced performance: Zambrano Chaves et al. (2025) employed GPT-4 (OpenAI, 2023) to refine training data by removing temporal comparisons, references to prior exams and unnecessary language variations, while Bannur et al. (2024) expanded the textual context to include indications,

technique and comparison, and the visual context by including lateral and prior frontal views. Despite these advances, these foundation models remain constrained by a core limitation of MLLMs: the reduction in attention values over image tokens as more tokens are generated (Favero et al., 2024; Chu et al., 2025).

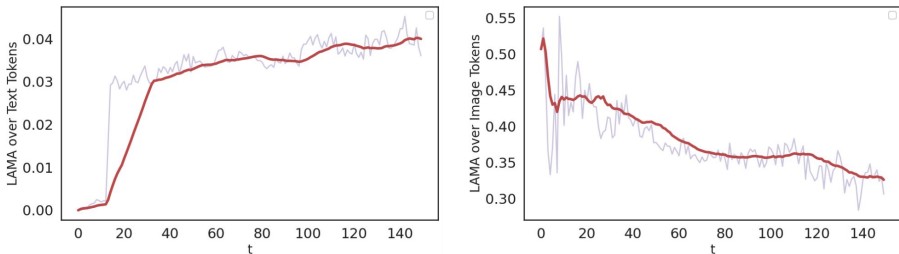

Figure 2: LAMA score calculated from 100 randomly sampled images from MIMIC-CXR dataset using LLaVA-Rad over text tokens (left) and image tokens (right). During the report generation process, we observe a pronounced decline in attention to image tokens accompanied by a steady increase in reliance on linguistic priors.

**Motivation.** We observe that, as report generation progresses, the model's attention increasingly relies on prior linguistic context rather than the image information. The maximum weight in multi-head attention layer (Vaswani et al., 2017) can be interpreted as a signal of the model's strong confidence in the corresponding input token (Voita et al., 2019; Huang et al., 2024). Based on this insight, we define *Layer-Averaged Max Attention (LAMA)*, which can be computed over any subset of target tokens $S$ (e.g., image tokens or generated text tokens). Let $A_t^{(l,h)} \in \mathbb{R}^N$ denote the attention weights for generated token $t$ in layer $l$ and head $h$. Then the LAMA score at step $t$ is:

$$\text{LAMA}_t(S) = \frac{1}{L} \sum_{l=1}^{L} \max_h \left( \sum_{i \in S} A_t^{(l,h)}[i] \right). \tag{1}$$

From the MIMIC-CXR (Johnson et al., 2019a) dataset, we compute $\text{LAMA}_t(S_{\text{vis}})$, where $S_{\text{vis}}$ denotes the set of all image tokens, for 100 randomly sampled X-rays from the test set. We observe a clear downward trend in $\text{LAMA}_t(S_{\text{vis}})$ over the generation steps (Fig. 2), suggesting a decay in attention to the image tokens during the generation process, accompanied by an increase in attention over the language priors. We hypothesize that this causes the model to learn spurious co-occurrences of pathology due to inherent biases in the training datasets. A typical example of such spurious pathology co-occurrence arises with cardiomegaly and pulmonary edema. In many cases, these two findings frequently appear together because both are associated with congestive heart failure (Siwik et al., 2023). As a result, when the model increasingly relies on textual priors, the presence of cardiomegaly alone serves as a language cue that strongly biases subsequent tokens toward the associated pathology (pulmonary edema in this case), even if the visual evidence is absent. Similarly, pleural effusion (fluid accumulation) can mechanically lead to some degree of rounded atelectasis (lung collapse) due to compression (Mancò et al., 2024). This statistical co-occurrence

can also lead the model to generate spurious findings simply because they commonly appear together in the training distribution, rather than being grounded in the underlying image evidence.

Given these observations, we introduce **Category-Wise Contrastive Decoding**, a novel and modular method that is designed to enhance *structured findings generation* in radiology foundation models. Category-Wise Contrastive Decoding aims to mitigate the problems of generating spurious co-occurrences and reduced attention on visual tokens with increase in output length in two ways: (i) Category-Specific Parametrization - We generate a findings report *category-wise* under eight anatomical headers, as defined by Delbrouck et al. (2025): Lungs and Airways, Pleura, Cardiovascular, Hila and Mediastinum, Tubes, Catheters, and Support Devices, Musculoskeletal and Chest Wall, Abdominal, and Other. Henceforth, we refer to these anatomical headers as categories of a structured radiology report. (ii) Masked Contrastive Decoding - An inference time strategy, where instead of normal greedy decoding, we sample from a contrasted distribution obtained by masking the X-ray using category-specific visual prompts. Introducing a contrastive objective at inference time prevents hallucinations arising from prior language bias learned during training.

## 2. Methods

**Vision Language Modeling.** Large language models (LLMs) process sequences of text tokens to generate textual output in an autoregressive manner. This mechanism can be extended to images by adding a vision encoder that extracts visual features, which are then projected into the text embedding space so they can be fed to the language model (LM) as additional input tokens. In practice, this is done by using a pre-trained vision backbone (e.g., ViT (Dosovitskiy et al., 2021) or a CNN-based encoder (Ge et al., 2024)) to extract a sequence of visual feature embeddings, which are then mapped into the language model's embedding space via a learnable multi-modal adapter (Li et al., 2023a; Alayrac et al., 2022), typically implemented as a multilayer perceptron (MLP). The resulting image tokens have the exact dimensions as input text tokens, allowing them to be concatenated to the LLM's input sequence (Liu et al., 2023). This unified token stream is then processed autoregressively by the LLM, enabling it to generate text conditioned both on the input image and text. This architecture serves as the foundation of an MLLM (Li et al., 2023a; Liu et al., 2023; Alayrac et al., 2022). Intuitively, this design allows images to be treated as a sequence of "visual words" that are compatible with text tokens. By projecting visual features into the same embedding space as text, the language model can jointly reason over both modalities using its standard autoregressive decoding mechanism.

Formally, consider a sample $(I, r)$ where $I$ represents a Chest X-ray image and $r$ represents the corresponding radiology findings report. Given an image encoder $E_{img}(\cdot)$, we obtain visual features $I' = E_{img}(I)$, which are then projected into the LM token embedding space by the multi-modal projector $\lambda(\cdot)$, we get $v = \lambda(I')$. The LM receives both the visual tokens and the text tokens as a single input sequence, usually with visual tokens provided first, followed by textual tokens. Let the textual input tokens be $u$. Then at step $t$, input to the model is $\chi_t = concat(v, u, r_{<t})$. The LM $\theta$ then processes this concatenated input sequence $\chi_t$ to give the hidden state $h_t = \theta(\chi_t)$ that is then passed to the LM head which projects $h_t$ from $d_m$ to $|V|$ to get logits $z_t = \theta_{head}(h_t)$, where $d_m$ is the LM's internal dimen-

sionality and $V$ is the vocabulary. Finally, we decode the findings reports auto-regressively from $P(r_t \mid \chi_t) = softmax(z_t)$. At each decoding step, the model predicts the next text token conditioned on the image, the textual prompt, and all previously generated tokens. The final generated report sequence is factorized as,

$$P_{(\theta,\lambda)}(r \mid v, u) = \prod_{t=1}^{|r|} P_{(\theta,\lambda)}(r_t \mid v, u, r_{<t}). \tag{2}$$

### 2.1. Category-Specific Parametrization

A free-text radiology findings report can be written as a structured findings report under eight categories (anatomical headers), as mentioned in appendix Sec. C. Foundational RRG models fine-tuned on an SRRG dataset are used to generate an SRR via a single continuous decoding process (Delbrouck et al., 2025). Based on the empirical observation described earlier (Fig. 2), we generate the findings report under each category in multiple *independent* forward passes to maintain visual grounding on the image tokens $v$ and reduce bias arising from excessive attention to previously generated tokens $r_{<t}$. By resetting the decoding context for each category, the model is encouraged to attend directly to the image rather than relying on textual priors from earlier sections.

Each structured findings report can be represented as $r = (r_{c_1}, r_{c_2}, \dots, r_{c_n})$ where, $1 \le n \le 8$, and $c_i$ represents a category $i$. As seen in Fig. 3, to specialize by category without disregarding the radiology priors of the base MLLM, we use low-rank adaptation (LoRA Hu et al. (2022)) on top of a base MLLM (Zambrano Chaves et al., 2025). This design enables category level specialization while preserving the general medical knowledge encoded in the base model. Given a foundation MLLM $\theta$ with weights $W$, we train $\Delta W = \Delta\theta_{c_i}$ for each category $c_i$, which decomposes into two low-rank weight matrices, significantly reducing the number of trained parameters. During inference, for every image $I$, we generate the category specific report $\tilde{r}_{c_i}$ using the MLLM $\theta + \Delta\theta_{c_i}$ (henceforth written as $\theta_{c_i}$) and category prompt $u_i$ for all $c_i$. We then concatenate $\tilde{r}_{c_i}$ from all categories to get the predicted structured report $\tilde{r} = (\tilde{r}_{c_1}, \tilde{r}_{c_2}, \dots, \tilde{r}_{c_n})$.

### 2.2. Category-Wise Contrastive Decoding for RRG

Traditionally, we sample from the distribution $P(y \mid c, x)$, where y is the output, x is the input, and c is the key context (e.g., an image) required to generate the relevant output. On the other hand, in Contrastive Decoding, we sample from the distribution obtained by contrasting $P(y \mid c, x)$ with $P(y \mid x)$. The distribution $P(y \mid x)$ can be thought of as representation of the model's prior bias, since it ignores the key context $c$. By contrasting these two distributions, we suppress continuations that are likely under this biased prior alone and amplify those whose probability increases when $c$ is taken into account, effectively encouraging the model to focus on context-relevant information and produce more accurate, grounded outputs.

Inspired by the contrastive decoding for natural images (Wan et al., 2025), we propose Category-Wise Contrastive Decoding (CWCD) for Radiology Report Generation. As seen in Fig. 3, given a chest X-ray $I$ and corresponding *category-specific* bounding boxes $b_{c_i}$, we mask all the pixels present within the regions covered by $b_{c_i}$ to get $I^b_{c_i} = mask(I, b_{c_i})$. We

then do two forward passes through $\theta_{c_i}$ to obtain $P(r_{c_i}^t \mid I_{c_i}, u_{c_i}, r_{c_i}^{<t})$ and $P(r_{c_i}^t \mid I_{c_i}^b, u_{c_i}, r_{c_i}^{<t})$ called the *base* and *masked* probabilities respectively. Specifically, we contrast the base and masked log-probabilities using a weighted difference to define a distribution over the next token:

$$CD(r_{c_i}^t) = \text{softmax}\Big[(1 + \alpha) \cdot \log P\big(r_{c_i}^t \mid I_{c_i}, u_{c_i}, r_{c_i}^{<t}\big) - \alpha \cdot \log P\big(r_{c_i}^t \mid I_{c_i}^b, u_{c_i}, r_{c_i}^{<t}\big)\Big]. \quad (3)$$

$$= \text{softmax}\Big[\log P\big(r_{c_i}^t \mid I_{c_i}, u_{c_i}, r_{c_i}^{<t}\big) + \alpha \log \frac{P\big(r_{c_i}^t \mid I_{c_i}, u_{c_i}, r_{c_i}^{<t}\big)}{P\big(r_{c_i}^t \mid I_{c_i}^b, u_{c_i}, r_{c_i}^{<t}\big)}\Big]. \quad (4)$$

This shows that CWCD starts from the base distribution and adds a contrastive term proportional to logarithm of the ratio between the base and masked probabilities, upweighting tokens whose probability increases when the category-specific region is visible and downweighting those that remain likely even when it is masked. The weighting factor $\alpha$ determines how strongly the contrast affects the selection: increasing $\alpha$ amplifies the emphasis on differences between the base and masked distributions. The next token $r_{c_i}^t$ is chosen greedily based on the $CD(\cdot)$ scores. This token is then appended to both the base and masked sequences to compute the probabilities for the subsequent timestep. By operating in log-probability space (Eq. 3), the method preserves meaningful contrast even for tokens with low probability.

## 2.3. Plausibility-Based Vocabulary Subselection

While Category-Based Contrastive Decoding effectively contrasts the base and masked distributions, applying it indiscriminately at every timestep can undesirably penalize tokens that both distributions assign high probability to. These are often common-sense tokens that satisfy basic grammatical or linguistic constraints, which can be generated even with a masked chest X-ray input. Such penalization can reduce the final probability of highly plausible tokens, potentially leading to unintended outputs. To address this, we employ a Plausibility-Based Vocabulary Subselection through an adaptive plausibility constraint, inspired by Li et al. (2023b).

At each decoding step, we truncate the candidate token set based on the unmasked log-probabilities: only tokens whose probability exceeds a fraction $\beta$ of the maximum probability token in the current step are retained for softmax after contrasting. This ensures highly probable and linguistically apparent tokens are preserved. In contrast, implausible or low-probability tokens are excluded, resulting in a subselected vocabulary at each timestep over which the contrastive softmax is computed:

$$V_{sub}^t = \{\forall r^t \in V : \log \text{P}\big(r^t \mid I, u, r^{<t}\big) \geq \max_{r^t} \beta \cdot \log \text{P}\big(r^t \mid I, u, r^{<t}\big)\}. \quad (5)$$

The overall category-based contrastive objective becomes:

$$CD(r_{c_i}^t) = \text{softmax}\left(\mathbb{I}(r_{c_i}^t) \cdot \log \frac{P(r_{c_i}^t \mid I_{c_i}, u_{c_i}, r_{c_i}^{<t})^{1+\alpha}}{P(r_{c_i}^t \mid I_{c_i}^b, u_{c_i}, r_{c_i}^{<t})^\alpha}\right), \quad (6)$$

$$\mathbb{I}(r_{c_i}^t) = \begin{cases} 1 & \text{if } r_{c_i}^t \in V_{\text{sub}}^t \\ -\infty & \text{otherwise.} \end{cases} \quad (7)$$

We use $\beta = 0.50$ (ablation study in Sec. F.1) and $\alpha = 1$ to balance the base and contrastive terms without overly suppressing plausible tokens, following Wan et al. (2025).

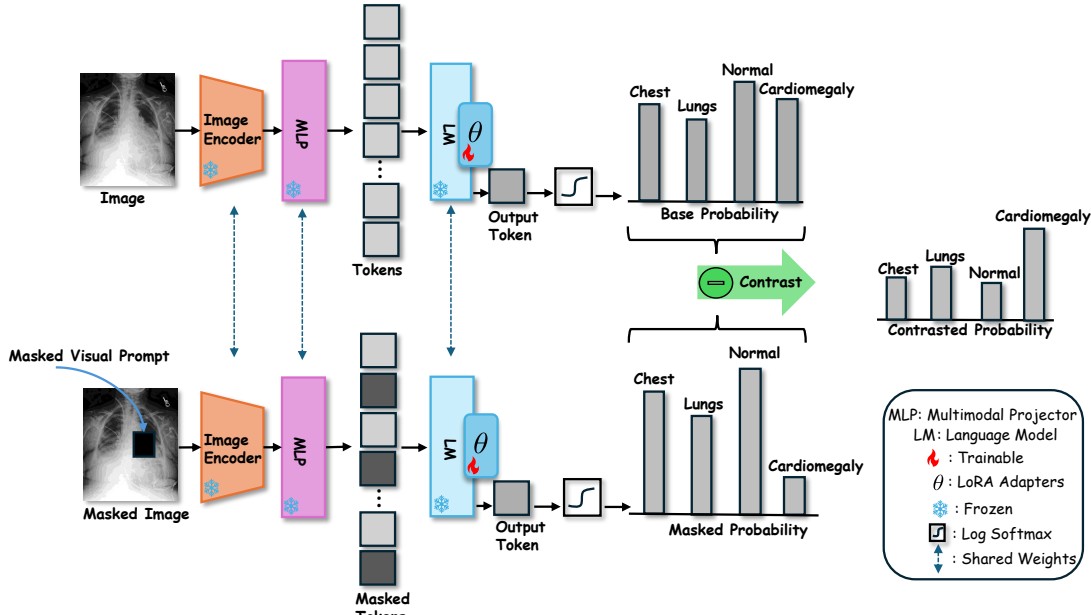

Figure 3: An overview of CWCD framework for the "Cardiovascular" Anatomical category. The base log probability distribution is contrasted with the masked log probability distribution using Eq.6. We then sample the highest probability token from the final distribution. This process repeats for each token in an auto-regressive form to obtain a Category report. Reports across all categories are aggregated to obtain a full structured report.

## 3. Experiments and Results

### 3.1. Datasets

For training category-wise adapters, we use X-rays from **MIMIC-CXR** (Johnson et al., 2019a,b) and we source the corresponding structured findings reports from **SRRG-Findings** (Delbrouck et al., 2025). To support category-wise parametrization, we parse each structured report and extract the bullet-point observations corresponding to every anatomical header, thereby constructing eight separate **category-specific datasets**. Each dataset contains all observations associated with its respective anatomical region to be used for training each category-wise adapter. For generating masks for CWCD, we use bounding-box annotations derived from the REFLACX dataset (Bigolin Lanfredi et al., 2022) and its derived dataset, LATTE-CXR (Ghelichkhan and Tasdizen, 2025).

**REFLACX** contains 3,032 readings corresponding to 2,616 unique chest radiographs. It provides radiologist eye-tracking data and manually drawn ellipses that indicate abnor-

mal findings, along with synchronized report transcriptions. **LATTE-CXR** repurposes the REFLACX annotations to generate bounding-box region annotations aligned with the sentences describing the abnormalities. For gaze-based pairs, radiologist fixations during report dictation are aggregated into gaussian heatmaps, processed to retain salient regions, and enclosed in axis-aligned rectangles to form bounding boxes aligned with each sentence. Expert-drawn ellipses from REFLACX are also converted into bounding boxes, providing explicit abnormality localization. These boxes represent regions attended to by radiologists rather than exact lesion boundaries. In total LATTE-CXR includes 13,751 gaze-based region–sentence pairs constructed from 2,742 MIMIC-CXR images. We follow the official MIMIC-CXR split and combine the test and validation sets to obtain a final test set of 912 X-rays. Category-specific bounding boxes are obtained by classifying each sentence–box pair into one of eight anatomical categories using DeepSeek (DeepSeek-AI, 2025).

Overall, we utilize frontal X-rays from MIMIC-CXR, structured findings reports from SRRG-Findings, and during inference, we employ category-specific bounding boxes from LATTE-CXR. Further details about the datasets can be found in appendix C.

### 3.2. Implementation Details

We use LLaVA-Rad (Zambrano Chaves et al., 2025) as our baseline MLLM model. LLaVA-Rad uses Vicuna-7b-v1.5 (Chiang et al., 2023) as the base language model and BioMedCLIP (Zhang et al., 2025a) as the image encoder, which is trained on large-scale multimodal biomedical data. For each of the eight categories, we train a rank-1 LoRA adapter, training ∼500k parameters per adapter. Across all categories, the total number of parameters trained is equivalent to those in a rank-8 adapter. We trained each adapter for one epoch on the corresponding category-specific dataset. All adapters are trained on a single 80GB A100 GPU. Each adapter takes between 4 and 16 hours, depending on the number of training samples in the category. We use a batch size of 48, a learning rate of 0.0001, and the AdamW (Loshchilov and Hutter, 2019) optimizer.

### 3.3. Evaluation Protocol

**Baselines.** We comprehensively evaluate against a diverse set of baseline radiology foundation models. All baseline models are pre-trained on the MIMIC-CXR dataset for generating free-text findings reports. Delbrouck et al. (2025) fine-tuned CheXpert-Plus (Hugging Face, 2025b), CheXagent-2 (Chen et al., 2024; Hugging Face, 2025a) and MAIRA-2 (Bannur et al., 2024; Hugging Face, 2025d) to generate SRR. Kang et al. (2025) fine-tuned Lingshu (Team et al., 2025; Hugging Face, 2025c) and MedGemma (Sellergren et al., 2025; Hugging Face, 2025e) to generate SRR. We trained LLaVA-Rad to generate SRR. CheXpert-Plus and CheXagent-2 were fully fine-tuned. For MAIRA-2 and LLaVA-Rad, rank 8 LoRA adapters were trained. For Lingshu and MedGemma, rank 32 LoRA adapters were trained.

**Metrics.** We evaluated the generated radiology reports using a combination of natural language generation (NLG) and clinical efficacy (CE) metrics, each capturing distinct aspects of report quality. For NLG, BLEU-1–4 (Papineni et al., 2002) measures n-gram overlap with reference reports, where lower-order BLEU (e.g., BLEU-1) emphasizes lexical precision and higher-order BLEU (e.g., BLEU-4) captures short phrase consistency. ROUGE-1,2,L

(Lin, 2004) focuses on recall, measuring how much of the reference content is covered, with ROUGE-L additionally reflecting structural similarity. BERTScore (BS) (Zhang* et al., 2020) evaluates semantic similarity using contextual embeddings, capturing meaning even when phrasing differs.

For clinical validity, F1-RadGraph (Jain et al., 2021; Delbrouck et al., 2022) evaluates the accuracy of entities (findings, anatomy) and relations, with simple, partial, and complete scores indicating varying levels of clinical precision. We measure the weighted average precision, recall, and F1 score over 55 SRR-BERT labels (Delbrouck et al., 2025), which enables more diverse evaluation compared to 14 CheXbert (Smit et al., 2020) disease labels.

### 3.4. Results

We evaluate the Category-Wise Contrastive Decoding (CWCD) framework on the Structured Radiology Report Generation (SRRG) task on the MIMIC-CXR derived test dataset, as defined in Sec. 3.1, against multiple state-of-the-art radiology foundation models. We conduct the SRRG evaluation in the same way as Delbrouck et al. (2025), except that we do not penalize the baseline models for not generating a category or generating an extra category; this results in overall higher baseline scores. CWCD demonstrates consistent improvements over all baseline models across both natural language generation and clinical efficacy metrics. In Table 1, CWCD achieves the highest score across all NLG metrics indicating more fluent, coherent, and semantically aligned report generation compared to the baselines.

Table 2 shows that CWCD also improves clinical validity, with F1RadGraph scores surpassing all other models. SRR-BERT metrics further confirm that CWCD generates clinically accurate findings with high precision (68.59) while maintaining competitive recall (61.08) and F1-Score (62.51). The higher precision indicates that CWCD produces fewer spurious or irrelevant findings, reducing the generation of pathology co-occurrences that are biased by language priors in the training data. The competitive recall shows that relevant findings are still captured, and the improved F1 suggests a better overall balance between accuracy and coverage. Taken together, the higher F1RadGraph scores, improved precision, and robust F1 indicate that CWCD enhances the overall clinical efficacy of generated reports while mitigating spurious correlations.

Table 1: Evaluation of CWCD versus Radiology Foundation Models on SRRG task on **NLG** Metrics defined in Sec. 3.3. Best scores are in **bold** and second best are underlined.

| Model | BL-1 | BL-2 | BL-3 | BL-4 | BS | R-1 | R-2 | R-L |
|---|---|---|---|---|---|---|---|---|
| CheXpert-Plus | 24.25 | 13.46 | 8.41 | 3.83 | 47.21 | 31.72 | 15.83 | 29.45 |
| MedGemma | 23.60 | 13.74 | 9.14 | 4.59 | 47.67 | 32.80 | 16.91 | 30.13 |
| Lingshu | 24.76 | 12.84 | 7.22 | 2.22 | 47.15 | 29.73 | 14.62 | 27.74 |
| CheXagent-2 | 23.35 | 13.71 | 8.80 | 4.59 | 48.03 | 32.79 | 16.84 | 30.28 |
| MAIRA-2 | 24.31 | 13.87 | 8.42 | 3.79 | 48.57 | 33.07 | 17.47 | 31.18 |
| LLaVA-Rad | 24.22 | 14.45 | 9.00 | 4.74 | 48.31 | 32.79 | 17.06 | 30.45 |
| CWCD | **27.76** | **16.77** | **11.53** | **6.60** | **50.22** | **35.26** | **20.25** | **33.27** |

Table 2: **Clinical Efficacy** Metrics as defined in Sec. 3.3.

| Model | F1Rad-S | F1Rad | F1Rad-C | Pr | Rc | F1 |
|---|---|---|---|---|---|---|
| CheXpert-Plus | 28.71 | 22.89 | 19.80 | 62.44 | 59.47 | 58.72 |
| MedGemma | 30.11 | 24.49 | 21.19 | 63.03 | 60.64 | 59.62 |
| Lingshu | 27.86 | 23.82 | 20.84 | 56.02 | 53.60 | 52.90 |
| CheXagent-2 | 30.27 | 24.29 | 21.11 | 64.20 | 60.74 | 60.67 |
| MAIRA-2 | 30.54 | 25.26 | 22.08 | 65.36 | 60.92 | 61.03 |
| LLaVA-Rad | 30.30 | 24.06 | 20.92 | 65.48 | **63.38** | 62.12 |
| CWCD | **32.96** | **27.96** | **24.60** | **68.59** | 61.08 | **62.51** |

## 3.5. Ablation Study

In this section, we conduct an ablation study to understand the contribution of each component in our approach. We perform a systematic ablation on the SRRG-Findings task using the dataset described in Sec. 3.1. Tab. 3 summarizes the results for six model variants, each incrementally adding or removing key mechanisms of the complete CWCD framework. Applying CD and vocabulary subselection (VS) to SRR yields modest gains (2nd row) across most metrics but also causes a notable drop in F1-SRR-BERT, indicating limited clinical reliability. Introducing Category-Wise parametrization (CW) yields substantial improvements (3rd row) across both NLG and CE metrics, demonstrating the effectiveness of reducing the number of generated tokens within a single set of forward passes. Masking all visual prompts (VP) in CWCD (5th row) further degrades performance, falling even below CW decoding. Similarly, removing VS from CWCD (4th row) results in a significant performance drop, highlighting the importance of filtering out low-probability tokens during CD. Overall, the complete framework, combining CW parametrization, VS, and category-specific VPs achieves the strongest performance across all metrics.

Table 3: Ablation study of CWCD on SRRG-Findings task on dataset defined in Sec. 3.1. VS stands for Vocabulary Subselection. VP stands for Visual Prompt. CW stands for Category-Wise report generation. Overall CWCD framework metrics are highlighted in green.

| Model | BL-4 | BS | R-1 | R-L | F1Rad | F1-SRR |
|---|---|---|---|---|---|---|
| LLaVA-Rad (Baseline) | 4.74 | 48.31 | 32.79 | 30.45 | 24.06 | 62.12 |
| LLaVA-Rad w/ CD+VS | 5.13 | 49.75 | 33.86 | 31.62 | 24.70 | 59.98 |
| CW | 6.46 | 49.58 | 34.83 | 32.91 | 27.31 | **62.57** |
| CWCD w/o VS | 6.23 | 49.77 | 34.15 | 32.00 | 26.53 | 60.40 |
| CWCD w/ all VP | 6.09 | 49.75 | 34.57 | 32.55 | 27.40 | 62.22 |
| CWCD w/ Cat-Spec. VP | **6.60** | **50.22** | **35.26** | **33.27** | **27.96** | 62.51 |

### 3.6. Out-of-Distribution Performance

We perform out-of-distribution (OOD) evaluation on the test split of IU-Xray (Demner-Fushman et al., 2016). Previously, while evaluating performance on the MIMIC-CXR dataset, we used ground truth visual prompt annotations from Latte-CXR. Given that no such annotations exist for IU-Xray, following Zhu et al. (2025); Wan et al. (2025), we use the Grounding DINO (Liu et al., 2024) model to extract visual prompts for each of the eight SRR categories. Further details about fine-tuning Grounding DINO for our use can be found in appendix Sec D.

Tables 4 and 5 show that CWCD demonstrates strong out-of-distribution generalization, consistently outperforming foundation models across both NLG and clinical efficacy metrics. While MedGemma also exhibits strong OOD performance, this may be partially attributable to its substantially larger fine-tuning capacity, as it employs rank-32 LoRA adapters, whereas CWCD is trained with parameters equivalent to a rank-8 adapter ($8 \times$ rank-1). Despite this disparity in adaptation capacity, CWCD achieves the best performance on 11 out of 14 metrics, highlighting the robustness of our method under distributional shift.

Table 4: Evaluation of CWCD on the out-of-distribution IU-Xray test set on NLG Metrics.

| Model | BL-1 | BL-2 | BL-3 | BL-4 | BS | R-1 | R-2 | R-L |
|---|---|---|---|---|---|---|---|---|
| CheXpert-Plus | 27.03 | 15.46 | 7.70 | 1.77 | 45.06 | 36.95 | 17.93 | 34.78 |
| MedGemma | 27.27 | 16.42 | 7.98 | 1.55 | **48.18** | 40.52 | 19.29 | 35.93 |
| Lingshu | 27.15 | 15.20 | 7.08 | **3.02** | 44.72 | 35.62 | 16.96 | 33.14 |
| CheXagent-2 | 26.30 | 14.67 | 8.01 | 1.70 | 45.24 | 37.24 | 18.26 | 34.44 |
| MAIRA-2 | 26.68 | 16.01 | 9.18 | 1.41 | 48.17 | 38.23 | 19.43 | 35.67 |
| LLaVA-Rad | 27.63 | 16.48 | 8.44 | 1.68 | 46.06 | 39.08 | 21.00 | 35.79 |
| CWCD | **28.47** | **17.49** | **9.83** | 2.00 | 47.81 | **40.63** | **22.76** | **37.53** |

Table 5: Evaluation on Clinical Efficacy Metrics.

| Model | F1Rad-S | F1Rad | F1Rad-C | Pr | Rc | F1 |
|---|---|---|---|---|---|---|
| CheXpert-Plus | 34.76 | 28.45 | 23.05 | 75.24 | 76.56 | 73.70 |
| MedGemma | 42.31 | 33.79 | 28.90 | 78.67 | **86.26** | 78.29 |
| Lingshu | 35.88 | 30.48 | 25.13 | 65.86 | 71.80 | 67.05 |
| CheXagent-2 | 34.22 | 28.59 | 22.35 | 81.67 | 81.26 | 78.71 |
| MAIRA-2 | 35.43 | 30.20 | 25.50 | 83.30 | 82.87 | 80.44 |
| LLaVA-Rad | 41.07 | 33.36 | 27.65 | 81.90 | 83.76 | 79.84 |
| CWCD | **42.76** | **35.73** | **29.03** | **89.15** | 82.63 | **83.79** |

**Limitations.** Although our training pipeline is relatively lightweight, inference remains computationally expensive: predictions must be generated across all eight categories, and the CD component requires two forward passes per token. As a result, the overall inference process is time-intensive. Additionally, because the structured reports were derived by refor-

mulating MIMIC-CXR free-text reports using a language model, there is a risk that subtle inconsistencies or biases may have been introduced by the model. Finally, our pipeline relies on automated anatomical classification by a large language model; while prior work shows strong performance (Tordjman et al., 2025; Niu et al., 2025), misclassification errors may propagate downstream and affect report generation quality.

## 4. Conclusion

Foundational radiology MLLMs generate a radiology report in a single set of forward passes. We show that this leads to reduced attention on image tokens and over-reliance on prior textual tokens leading to limited clinical accuracy of automated reports. To address these issues, we introduce Category-Wise Contrastive Decoding (CWCD), a framework that generates category-wise structured reports through category-specific parameterization and masked contrastive decoding. Experiments on MIMIC-CXR and the out-of-distribution IU-Xray demonstrate that CWCD strengthens visual grounding, enhances clinical fidelity, and improves the linguistic quality of generated reports, advancing the capabilities of foundational radiology MLLMs.

**Acknowledgment.** This work was supported by the US NSF CAREER award IIS-2239537.

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

# Appendix A. Extended Motivation

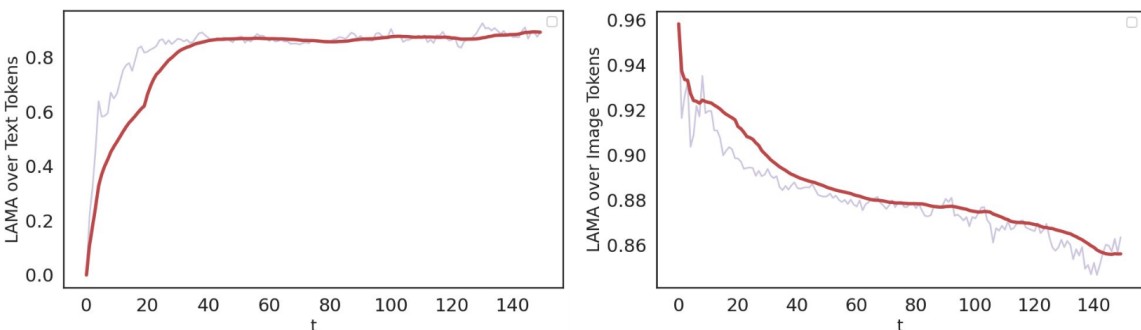

Figure 4: We replicated the experiment presented in Sec. 1 on CheXagent-2 to demonstrate that the problem of attention decay over image tokens during token generation also affects other MLLMs.

# Appendix B. Related Work

**Structured Findings Generation.** Findings section of a radiology report is comprised of visual observations from a given chest X-ray. Usually, these are free-text reports but there is a growing body of work that establishes the utility of structured reports. Marcovici and Taylor (2014) showed that clinicians rated structured reports to be significantly more complete and more effective. Buckley et al. (2018) showed that structured reports allowed better recall of diagnosis and critical findings and overall both referring physicians and radiologists preferred structured reports over free-text reports (Jorg et al., 2023). Recently, Delbrouck et al. (2025) introduced a desiderata for structured reporting where they divided the entire radiology report into predefined sections and within the findings section, they further divided by 8 anatomical headers mentioned previously. They converted the free-text reports of MIMIC-CXR and CheXpert Plus to structured reports and introduced two new datasets called SRRG-Findings and SRRG-Impression. Kang et al. (2025) further added clinical context like multiple views, clinical indication, imaging techniques used and prior studies to give a new dataset called contextualized SRRG (C-SRRG).

Beyond clinical utility, in automated report generation systems, structured reports help mitigate distributional shift between textual reports originating from different datasets, where the same clinical finding may be described in markedly different styles due to linguistic, institutional, or regional differences among radiologists. By standardizing both the reporting categories and the linguistic style, structured reports reduce this variability and provide more consistent supervision for model training. Additionally, the natural division of the findings section into well-defined anatomical categories enables category-wise parametrization and modular report generation. We believe this structure promotes stronger visual grounding by preventing over-reliance on language priors and by reducing the number of tokens generated within each continuous forward pass.

**Contrastive Decoding.** Contrastive decoding (CD) is a training-free inference time strategy for reducing hallucinations in text generative models (Li et al., 2023c; Leng et al., 2024; O'Brien and Lewis, 2023). The main idea of CD is to overcome statistical biases (like object co-occurrences) inherent in the training data and in case of MLLMs, prevent over-reliance on textual priors learned during the pre-training of the LLM. Contrasting with the distribution produced after masking the key information required to generate the correct output penalizes the tokens that are generated when the key information is missing, effectively exposes the prior bias of the model. Various approaches for CD in MLLMs have been tried, Leng et al. (2024) contrast output distributions derived from original and distorted visual inputs, Xu et al. (2025) contrast inter-layer representations, Wan et al. (2025) contrast model outputs produced with and without visual prompts. While CD has worked well for mitigating hallucinations in natural image captioning tasks, its use for medical tasks has been very limited. Xu et al. (2024) developed Alternative CD for medical information extraction task, where they alternately contrasted output distributions from sub-task modules. Zhang et al. (2025b) introduces a dual-stage CD mechanism for RRG. Both Xu et al. (2024) and Zhang et al. (2025b) contrast with text based approaches, whereas, to the best of our knowledge, we are the first to introduce an image based CD approach for RRG i.e., the contrasted distribution is generated by masking the X-ray instead of masking the text.

## Appendix C. Datasets

**MIMIC-CXR** dataset is a large publicly available collection of de-identified chest radiographs and accompanying free-text radiology reports. The dataset was sourced from the Beth Israel Deaconess Medical Center (BIDMC) in Boston, USA, and includes imaging studies collected as part of routine clinical care between 2011 and 2016. It contains $377,110$ chest X-ray images corresponding to $277,827$ imaging studies from $65,379$ patients. Most studies include both frontal (anteroposterior or posteroanterior) and lateral views, and the original images are stored in DICOM format. We use the JPEG format images provided in MIMIC-CXR-JPG (Johnson et al., 2019c).

All images in the dataset were acquired as part of routine clinical care using standard radiography equipment in a hospital environment and were subsequently de-identified in accordance with HIPAA regulations. The dataset was not curated for specific diseases; instead, it preserves the natural distribution of thoracic conditions and imaging characteristics encountered in real-world clinical practice. As a result, the images exhibit substantial clinical variability, including differences in patient positioning (e.g., anteroposterior and posteroanterior views), acquisition settings, image quality, and the presence of medical devices. The accompanying radiology reports were produced by board-certified radiologists at the time of image acquisition and are temporally aligned with the imaging studies.

**SRRG-Findings** dataset is derived from the findings section of reports in MIMIC-CXR and Chexpert-Plus (Chambon et al., 2024), which are converted into a standardized structured format using GPT-4 (OpenAI, 2023) following a strict set of desiderata. In SRRG, each free-text findings section is reorganized under a fixed set of anatomical headers: Lungs and Airways, Pleura, Cardiovascular, Hila and Mediastinum, Tubes, Catheters and Support Devices, Musculoskeletal and Chest Wall, Abdomen, and Other. Within each category, ob-

servations are expressed as bullet-point statements.

**IU-Xray** dataset from Indiana University is a publicly available chest X-ray dataset comprising 8,121 chest X-ray images and 3,996 associated radiology reports, collected from the picture archiving systems of the Indiana Network for Patient Care. The images and reports were de-identified automatically and then manually verified in accordance with HIPAA guidelines. For our evaluation, we randomly select 20% of the data as the test set, following previous work (Chen et al., 2022).

## Appendix D. Grounding DINO Fine-Tuning

Grounding DINO is an open-set object detector that takes an image and a text prompt as input and outputs bounding boxes corresponding to the specified text. While it demonstrates strong performance on natural images, we fine-tune Grounding DINO on LATTE-CXR to extract category-specific bounding boxes aligned with our anatomical headers.

As described in Sec. 3.1, LATTE-CXR contains 13,751 sentence–bounding box pairs. Each sentence–box pair is classified into one of eight anatomical categories using DeepSeek. The training set consists of 8,850 bounding box–anatomical region pairs, which are used to fine-tune Grounding DINO.

During fine-tuning, we optimize a contrastive loss (Radford et al., 2021) between object features and text tokens for classification, along with L1 and GIoU (Rezatofighi et al., 2019) losses for bounding box regression.

During inference for a given anatomical category, we input the chest X-ray and the corresponding anatomical header, and the model returns one or more relevant bounding boxes.

## Appendix E. Using Visual Prompts

In this section, we study the role of visual prompts (VPs) in our framework. While VPs have been used in prior work to enhance medical visual question answering (MedVQA) (Zhu et al., 2025) and zero-shot classification (Denner et al., 2025), to the best of our knowledge, no prior study has leveraged VPs in a training-free manner specifically to improve radiology report generation.

Since CWCD employs masked VPs during evaluation, we ensure a fair comparison by providing the baseline LLaVA-Rad model with VPs in two ways: (i) $\alpha$ blended visual prompts on the input X-ray, following prior work (Zhu et al., 2025; Denner et al., 2025), and (ii) masked VPs for contrastive decoding combined with vocabulary subselection (VS), effectively extending the approach of Wan et al. (2025) with VS.

As shown in Tab. 6, both approaches (rows 2 and 3) perform worse than category-wise report generation (CW, row 4), where no VPs are provided. We hypothesize that the $\alpha$ blended VP approach is less effective for radiology report generation than for MedVQA or zero-shot classification due to the open-ended nature of the task and the larger number of visual prompts per X-ray (4–5 vs. 1–2 in MedVQA).

Overall, these results suggest that addressing the fundamental issue of attention decay in MLLMs through category-wise report generation provides the largest performance gains, while the inclusion of masked VPs offers modest additional improvements.

Table 6: Ablation study of CWCD on dataset defined in Sec. 3.1 using ground truth VPs from LATTE-CXR. VS stands for Vocabulary Subselection. VP stands for Visual Prompt. CW stands for Category-Wise report generation.

| Model | VP | BL-4 | BS | R-L | F1Rad | F1 |
|---|---|---|---|---|---|---|
| LLaVA-Rad (Baseline) | No | 4.74 | 48.31 | 30.45 | 24.06 | 62.12 |
| LLaVA-Rad ($\alpha$ blended VP) | Yes | 3.34 | 44.33 | 27.30 | 19.22 | 49.15 |
| LLaVA-Rad (CD+VS) | Masked | 5.13 | _49.75_ | 31.62 | 24.70 | 59.98 |
| CW | No | _6.46_ | 49.58 | _32.91_ | _27.31_ | **62.57** |
| CWCD | Masked | **6.60** | **50.22** | **33.27** | **27.96** | _62.51_ |

## Appendix F. The Masking Mechanism

While generating a structured radiology report for a particular category, all pixels on and within the corresponding bounding boxes are blacked out (RGB value of 0,0,0), effectively removing the underlying visual information from the input image, as shown in Fig. 1. As a result, the MLLM generates tokens conditioned only on the remaining regions of the X-ray and the previously generated text tokens.

This masking mechanism is critical for contrastive decoding, as it enables a controlled comparison between tokens produced with and without access to the relevant visual region. By fully removing category-specific visual evidence, differences in the resulting outputs reflect the model's reliance on that region for generating category-specific descriptions. Partial masking or soft attenuation may allow residual visual cues to persist, weakening the contrastive signal. Therefore, complete masking provides a clear intervention for isolating the contribution of the masked region to the generated text.

### F.1. Hyperparameter Tuning

We analyze the effect of the vocabulary threshold hyperparameter $\beta$, which controls the minimum log-probability cutoff relative to the highest-probability token at each decoding step (Eq. 5). Tables 7 and 8 show the impact of varying $\beta$ on NLG and clinical efficacy metrics, with the baseline without Vocabulary Subselection highlighted in red and the chosen $\beta$ in green.

Very low values of $\beta$ (0.00–0.01), corresponding to minimal filtering, lead to lower overall performance in both NLG and clinical metrics, indicating that including low-probability tokens increases the risk of generating irrelevant or spurious content. Moderate values of $\beta$ (0.10–0.50) show steady improvements, with $\beta = 0.50$ achieving the best balance and strongest overall performance. Higher thresholds (0.75–0.90) maintain competitive results but offer limited additional gains and may slightly restrict the generation of relevant content.

Overall, these trends demonstrate that vocabulary subselection is a critical component of CWCD, and that an appropriately chosen $\beta$ effectively balances linguistic quality with clinical correctness.

Table 7: Effect of the hyperparameter $\beta$ (Eq. 5) on CWCD's overall performance on **NLG** metrics. $\beta$ used in CWCD is highlighted in green and the baseline without Vocabulary Subselection is highlighted in red.

| $\beta$ | BL-1 | BL-2 | BL-3 | BL-4 | BS | R-1 | R-2 | R-L |
|---|---|---|---|---|---|---|---|---|
| 0.00 | 27.15 | 16.11 | 10.80 | 6.23 | 49.77 | 34.15 | 19.27 | 32.00 |
| 0.01 | 27.21 | 16.15 | 10.84 | 6.25 | 49.79 | 34.19 | 19.29 | 32.05 |
| 0.10 | 27.63 | 16.56 | 11.14 | 6.37 | 50.22 | 34.82 | 19.86 | 32.69 |
| 0.25 | 27.66 | 16.57 | 11.20 | 6.39 | 50.18 | 35.00 | 20.02 | 32.95 |
| 0.50 | **27.76** | **16.77** | **11.53** | **6.60** | **50.22** | **35.26** | 20.25 | **33.27** |
| 0.75 | 27.40 | 16.43 | 11.39 | 6.52 | 49.82 | 35.05 | **20.26** | 33.10 |
| 0.90 | 27.22 | 16.34 | 11.34 | 6.44 | 49.64 | 34.89 | 20.22 | 32.96 |

Table 8: Clinical Efficacy Metrics.

| $\beta$ | F1Rad-S | F1Rad | F1Rad-C | Pr | Rc | F1 |
|---|---|---|---|---|---|---|
| 0.00 | 31.21 | 26.53 | 23.22 | 65.53 | 59.76 | 60.40 |
| 0.01 | 31.30 | 26.62 | 23.31 | 65.61 | 59.78 | 60.46 |
| 0.10 | 31.96 | 27.25 | 23.90 | 66.79 | 60.23 | 61.33 |
| 0.25 | 32.30 | 27.45 | 24.06 | 67.68 | 60.68 | 61.97 |
| 0.50 | **32.96** | **27.96** | **24.60** | 68.59 | 61.08 | 62.51 |
| 0.75 | 32.34 | 27.49 | 24.23 | 68.75 | **61.21** | **62.68** |
| 0.90 | 32.23 | 27.41 | 24.08 | **68.76** | 61.07 | 62.58 |

