# OpenReview forum: "CWCD: Category-Wise Contrastive Decoding for Structured Medical Report Generation"
_MIDL.io/2026/Conference — MIDL 2026 Poster_

### Official Review · Reviewer_S4Tp · 2026-01-09

**Confidence:** 4
**Preliminary Rating:** 3
**Final Rating:** 5

**Summary:**

The paper proposes a method to improve structured radiology report generation with VLMs by (i) introducing decoupled LoRA weights for each of the eight pre-defined anatomical headers (categories), and (ii) by contrasting the full model’s predictions with those obtained under partially occluded visual context to further mitigate hallucinations. The proposed approach outperforms previous baselines in terms of both lexical and clinical metrics on the MIMIC-CXR derived test set.

**Strengths:**

- The problem of hallucinations in VLMs used for radiology report generation is clearly motivated. The authors also use a convincing example (in pg. 3) to demonstrate this in practice.

- The paper is well organized.

- The proposed method outperforms strong VLM baselines in terms of natural language generation as well as clinical metrics.

**Weaknesses:**

- The proposed approach is not entirely novel. More specifically, the finding that "the model's reliance on input image tokens drops as report generation progresses" is also shown in (Favero et al., 2024; Chu et al., 2025), while the proposed contrastive decoding follows (Wan et al., 2025). The application to structured radiology report generation is the novel aspect here.

- The proposed method relies on ground truth bounding boxes for each anatomical area, which might not be available in real-world practice.

**Detailed Comments:**

- In the Motivation subsection (pg. 3), the authors did not mention which VLM they used. Also, is this empirical finding consistent with other VLM architectures? (e.g., LLaVA-RAD, MAIRA-2 etc.)

- In the introduction of Section 2 ("Methods", towards the end of pg. 4) and in Section 2.1 (pg. 5), the main text is fairly hard to follow, especially for readers not familiar with VLMs.

- I believe that referring to anatomical headers as *categories* might be slightly confusing (those *categories* could be mistaken for pathologies instead).

**Justification Of Final Rating:**

The authors have significantly strengthened the paper during the rebuttal period. More specifically, the new experiments provided to support the main results, for example:

- the justification for the category-specific LoRA adapters,
- the LAMA values of CheXagent-2 that further support the motivation of this work (Appendix A), and
- the reported results on an OOD dataset (IU-Xray) with silver-standard bounding boxes (predicted by the Grounding-DINO model)

have effectively addressed my concerns. Therefore, I am happy to raise my score to 5 (Strong accept).

**Justification Of The Preliminary Rating:**

The proposed method yields performance gains in structured radiology report generation at the expense of a slightly higher computational cost (category-specific LoRA adapters). Also, for the contrastive decoding part, the model requires access to bounding boxes to mask the related anatomical areas, which might not be available in practice.

**Questions To Address In The Rebuttal:**

- Why does the model require category-specific LoRA adapters to perform well? This design choice increases the computational cost linearly with the number of categories. Instead, can the model be prompted in a different way for each anatomical header?

- Did the authors perform an ablation study to justify the choice of rank-1 LoRA adapters?

- In Section 3.1 (pg. 7), the authors mention the following:
> “Category-specific bounding boxes are obtained by classifying each sentence–box pair into one of eight anatomical categories using DeepSeek (DeepSeek-AI, 2025)”.

Does this process introduce any errors?

---

> ### Author Response · Authors · 2026-01-25
>
> We would like to thank the reviewer for their detailed comments and suggestions. Below, we address each concern and highlight the improvements in the manuscript in response to their comments.
>
> ---
>
> **Concerns regarding novelty.**
>
> **Summary.** We propose a fundamentally new framework for structured radiology report generation that mitigates attention decay in MLLMs via category-specific parametrization, offering a computationally lightweight alternative to prior reinforcement learning based approaches (Favero et al., 2024; Chu et al., 2025). We show that existing visual prompting and contrastive region guidance (Wan et al. 2025) methods do not improve report generation (Appendix E). Our method uniquely integrates category-wise parametrization with contrastive decoding guided by category-specific masked visual prompts, yielding substantial gains over existing baselines.
>
> We have further discussed nuanced aspects regarding novelty in detail below:
>
> 1. **Category-specific parametrization for SRRG:** We introduce a fundamentally different approach to structured radiology report generation (SRRG) by decomposing the task into multiple subtasks through category-specific parametrization, as opposed to the current approach of generating the entire SRR in a single forward pass, which leads to performance degradation due to attention decay. As illustrated in Figures 1 and 2.
>
> 2. **Use of visual prompts in SRRG:** While visual prompts have been employed to enhance medical visual question answering and zero-shot classification performance (e.g., Zhu et al., Denner et al.), to the best of our knowledge, prior work has not leveraged visual prompts in a training free manner to improve radiology report generation. We add an ablation study (appendix E) which demonstrates that neither simple visual prompting nor directly applying contrastive region guidance (Wan et al., 2025) improves report generation performance. Even after augmenting contrastive region guidance with vocabulary subselection (Table 6, 3rd row), its performance remains below CW decoding (4th row).
>
> 3. **Novel integration of category-wise parametrization with contrastive decoding:** In summary, our framework CWCD, which combines category-specific parametrization with contrastive decoding guided by category-specific masked visual prompts represents a fundamentally new approach for structured radiology report generation, distinct from prior radiology report generation methods.
>
> 4. **Addressing attention decay differently from prior work:** We acknowledge that previous studies (Favero et al., 2024; Chu et al., 2025) have observed that a model’s reliance on input image tokens decreases as report generation progresses. However, our solution differs fundamentally: while these works rely on reinforcement learning approaches that are computationally expensive training-wise and often challenging to converge, our category-specific parametrization provides a simpler alternative in the context of radiology report generation.
>
> ---
>
> **Reliance on ground truth bounding boxes.**
>
> We acknowledge that ground truth bounding boxes are often unavailable in practice. To address this, we fine-tuned a Grounding DINO based visual prompt generator to produce category-specific bounding boxes for X-rays, trained on MIMIC-CXR. This fine-tuned model was then used to generate visual prompts when evaluating CWCD on an out-of-distribution dataset (IU-Xray), where ground truth bounding boxes were not available. Our model achieved strong results (Sec. 3.6), effectively removing the reliance on ground truth annotations.
>
> ---
>
> **Detailed Comment 1:** We used the LAMA values from LLaVA-Rad in Fig. 2. To verify that this empirical observation holds across other models, we repeated the experiment on CheXagent-2 (based on the Phi-2 base model) in appendix A and observed similar results. We did not use MAIRA-2 here, as it is also based on LLaVA.
>
> **Detailed Comment 2:** We have briefly expanded these sections to enhance readability.
>
> **Detailed Comment 3:** We acknowledge that referring to anatomical headers as “categories” may be slightly confusing. To clarify, we have added a sentence in the last paragraph of the Introduction explaining this terminology, and we now include “(anatomical headers)” in parentheses at the first instance of “categories” in Sec. 2.1.

---

> > ### Author Response · Authors · 2026-01-25
> >
> > **Why category-specific LoRA adapters? Can different prompts for each anatomical header work instead?**
> >
> > Category-specific LoRA adapters are preferred because a single multi-task LoRA adapter forces all anatomical categories to share the same low-rank parameter subspace, which leads to gradient interference and representational compromise during training. This reduced task specific performance in multi-task learning even when tasks are clearly identified via prompts has been identified in previous works as well (Yang et al., 2024, Yang et al 2026). To further illustrate this effect, we compare category-wise report generation using two rank-1 LoRA adapters against a single rank-2 LoRA adapter, trained jointly on the categories “Lungs and Airways” and “Tubes, Catheters and Support Devices”. As shown in the Table below, the category-specific adapters substantially outperform the shared adapter, demonstrating the benefit of generating reports for each anatomical header through category-specific parameterization and not prompt guidance.
> >
> > |                     | BERT Score | F1RadGraph | F1-SRR |
> > |---------------------|------------|------------|--------|
> > | Single Rank 2 LoRA Adapter | 34.28      | 15.22      | 40.05  |
> > | Two Rank 1 LoRA Adapters   | **43.70**      | **20.67**      | **62.24**  |
> >
> > ---
> >
> > **Why rank 1 LoRA adapters?**
> >
> > Rank-1 LoRA adapters provide a simple yet effective baseline for comparison with a single rank-8 LoRA adapter (as used in LLaVA-Rad and MAIRA-2), since both configurations have the same total number of parameters, that is, a rank-1 adapter for each of the 8 categories equals the parameter count of a single rank-8 adapter. Rank-1 adapters establish a minimum performance baseline for CWCD at a relatively low training cost, while performance can be further improved by increasing the number of trainable parameters through higher-rank adapters.
> >
> > ---
> >
> > **Are any errors introduced by using DeepSeek for classification?**
> >
> > DeepSeek’s medical reasoning and classification performance has been thoroughly benchmarked in prior work (Tordjman et al., 2025), demonstrating performance comparable to other leading LLMs. Additionally, DeepSeek has shown results on par with radiologists in the evaluation of radiology reports (Niu et al., 2025), indicating strong clinical reasoning capabilities. In our experiments on an out-of-distribution dataset, visual prompts were generated using Grounding DINO trained on the outputs of DeepSeek classification (details in the Appendix D.). CWCD exhibited strong performance over foundation RRG models, suggesting that any errors introduced by DeepSeek’s classification were minimal. Nevertheless, we acknowledge that any automated classification process may introduce errors, and we have added this point to the limitations of our method.

---

> > > ### Author Response · Authors · 2026-01-25
> > >
> > > **References**
> > >
> > > - Alessandro Favero, Luca Zancato, Matthew Trager, Siddharth Choudhary, Pramuditha Perera, Alessandro Achille, Ashwin Swaminathan, and Stefano Soatto. *Multi-modal hallucination control by visual information grounding*. In Proceedings of the IEEE/CVF Conference on Computer Vision and Pattern Recognition (CVPR), pages 14303–14312, June 2024.
> > >
> > > - Xu Chu, Xinrong Chen, Guanyu Wang, Zhijie Tan, Kui Huang, Wenyu Lv, Tong Mo, and Weiping Li. *Qwen look again: Guiding vision-language reasoning models to re-attention visual information*, 2025.
> > >
> > > - David Wan, Jaemin Cho, Elias Stengel-Eskin, and Mohit Bansal. *Contrastive region guidance: Improving grounding in vision-language models without training*. In Computer Vision – ECCV 2024, pages 198–215, Cham, 2025. Springer Nature Switzerland.
> > >
> > > - Kangyu Zhu, Ziyuan Qin, Huahui Yi, Zekun Jiang, Qicheng Lao, Shaoting Zhang, and Kang Li. *Guiding Medical Vision-Language Models with Diverse Visual Prompts: Framework Design and Comprehensive Exploration of Prompt Variations*. In Proceedings of the 2025 Conference of the Nations of the Americas Chapter of the Association for Computational Linguistics: Human Language Technologies (Volume 1: Long Papers), pages 11726–11739, Albuquerque, New Mexico. Association for Computational Linguistics, 2025.
> > >
> > > - Denner, Stefan, et al. *Visual Prompt Engineering for Vision Language Models in Radiology*. MIDL 2025.
> > >
> > > - Yaming Yang, Dilxat Muhtar, Yelong Shen, Yuefeng Zhan, Jianfeng Liu, Yujing Wang, Hao Sun, Denvy Deng, Feng Sun, Qi Zhang, Weizhu Chen, and Yunhai Tong. *MTL-LoRA: Low-rank adaptation for multi-task learning*, 2024.
> > >
> > > - Ziyu Yang, Guibin Chen, Yuxin Yang, Aoxiong Zeng, and Xiangquan Yang. *Disentangling task conflicts in multi-task LoRA via orthogonal gradient projection*, 2026.
> > >
> > > - Tordjman, M., Liu, Z., Yuce, M., et al. *Comparative benchmarking of the DeepSeek large language model on medical tasks and clinical reasoning*. Nat Med 31, 2550–2555 (2025). https://doi.org/10.1038/s41591-025-03726-3
> > >
> > > - Niu S, Liu X, Huang L, Li Y, Wang G. *DeepSeek-R1 for automated scoring in radiology residency examinations: an agreement and test-retest reliability study*. BMC Med Educ. 2025 Nov 11;25(1):1581. doi: 10.1186/s12909-025-08184-6. PMID: 41219771; PMCID: PMC12607140.

---

### Official Review · Reviewer_uBUw · 2026-01-09

**Confidence:** 1
**Preliminary Rating:** 4
**Final Rating:** 5

**Summary:**

Category-Wise Contrastive Decoding (CWCD) is aimed at increasing clinical efficency and assist with automatic radiology report generation. Authors have aimed to improve the focus of the algorithm on images rather than inherent reliance of textual information. Evaluation metrics and well-defined ablations studies are highlights of this study.

**Strengths:**

- Problem is clearly defined and the technical limitations of multi-modal large language models (MLLMs) is highlited.
- Category-specific parameterization and a contrastive mechanism using normal versus masked X-rays is innovative.
- Utiliation of language generation (NLG) and clinical efficacy (CE) metrics allows for detailed evaluation of generated multimodal reports.

**Weaknesses:**

- Quantiative comparision or method-based imporvements not highlited in abstract.
- Disease focus or specific type of X-rays and other details are missing.
- Screening quality or basis of selection of X-ray images are not provided.

**Detailed Comments:**

- The study is focussed around MIMIC dataset. Was the algorithm evaluated with an external data source for better validation? perhaps from a different institution?
- Category vs. contrast - What is the importance of each component here? Would the algorithm perform better/worse if the images are from different levels of contrast?

**Justification Of Final Rating:**

The authors diligently responded to all the comments and updated the manuscript wherever necessary. They have demonstrated that their algorithm works across institutions and that is an important contribution above all. The critical comments were addressed and not further comments from my side.

**Justification Of The Preliminary Rating:**

Approach is innovative and a technoligical advancement that is requried (soon). Robustness of algorithm is there, but it needs validation with external data sources. The inclusion of ablation studies enchance the methological approach of this study.

**Questions To Address In The Rebuttal:**

- Would the algorithm work for contrast varied scans? Would it fare better or worse?
- How about anamolies like implanted devices? Would that influence the quality of generated report?
- Does the amount of text input influence the overall generated report quality? For example, if the information provided about the scan is limited to 20 words vs. 200 words - which case would see a better generated report?

---

> ### Author Response · Authors · 2026-01-25
>
> We thank the reviewer for their careful evaluation and thoughtful suggestions. Below, we address each comment and outline how the manuscript has been updated in response.
>
> ---
>
> **1. Further validation with an additional dataset is required.**
> We thank the reviewer for this suggestion. To evaluate the generalizability of our approach beyond the MIMIC dataset, we conducted an additional experiment using the IU-Xray dataset, which originates from a different institution. This allowed us to assess performance on an out-of-distribution dataset. The results (Sec. 3.6) demonstrate that our method maintains strong performance, indicating that CWCD generalizes well across datasets from different institutions.
>
> **2. Details about the X-ray dataset.**
> We have re-written the description of the MIMIC-CXR dataset with additional information. We would like to highlight that the dataset was not curated for specific diseases; instead, it preserves the natural distribution of thoracic conditions and imaging characteristics encountered in real clinical settings.
>
> **3. Would the algorithm work for contrast varied scans?**
> Although we did not perform a controlled experiment on image contrast, MIMIC-CXR and IU-Xray datasets were collected independently under different clinical imaging pipelines, with inherent differences in exposure, contrast, patient positioning, and acquisition protocols noted in their respective construction. The fact that CWCD exhibits strong performance across these two datasets, each exhibiting natural variability in image appearance, suggests that the method is robust to variations in contrast and imaging characteristics encountered in actual clinical settings.
>
> **4. How about anomalies like implanted devices?**
> Implanted devices come under the “Tubes, Catheters and Support Devices” category of the structured radiology report. We evaluated model performance for this category and observed substantial improvements with CWCD over the baseline(LLaVA-Rad):
>
> | Model    | BERT Score | F1Radgraph | F1-SRR |
> |----------|------------|----------|--------|
> | Baseline | 47.17      | 20.23    | 77.83  |
> | CWCD     | **51.32**      | **22.27**    | **82.89**  |
>
> **5. Does the amount of text input about the scan affect overall generated report quality?**
> The proposed method does not use any additional textual information about the scans as input.

---

### Official Review · Reviewer_YcHF · 2026-01-10

**Confidence:** 4
**Preliminary Rating:** 2
**Final Rating:** 4

**Summary:**

The paper addresses a critical limitation in current VLMs used for radiology report generation. where there is a tendency for model to generate "hallucinations" or spurious pathology co-occurrences due to a decay in visual attention as the generated text length increases. The authors introduce Category-Wise Contrastive Decoding (CWCD) to enhance Structured Radiology Report Generation.

**Strengths:**

* In general, the problem is clear and relevant. The pipeline is clearly structured in Figures 1, 2, and 3, showing significant improvement of  performance over prior multimodal baselines.
* The motivation of the paper is convincing. The model’s attention to image tokens drops significantly while its reliance on text tokens increases, effectively justifying the need for a mechanism that enforces visual grounding throughout the generation process.

**Weaknesses:**

My major concern relates to the category-specific bounding boxes.  The paper states that 'during inference, we employ category-specific bounding boxes from LATTE-CXR'. Since LATTE-CXR contains ground-truth annotations, this implies the proposed method utilizes lesion localization information during testing. Unless the baselines were also provided with these ground-truth regions as prior, this comparison appears to be unfair. The baselines are tasked with simultaneously localizing and describing pathologies, whereas CWCD is given the localization for free.

**Detailed Comments:**

* The detailed of how the bounding boxes were generated could be provided.

**Justification Of Final Rating:**

I would like to thank the authors for thoroughly addressing my concern regarding the mechanism of the bounding box by providing additional information in the manuscript. I am happy to increase the final rating accordingly.

**Justification Of The Preliminary Rating:**

The paper proposes a Category-Wise Contrastive Decoding (CWCD) framework to address hallucinated findings caused by the “attention decay” problem in long-sequence automated radiology report generation. However, a key component of the framework (the masking strategy) requires further discussion and clearer explanation.

**Questions To Address In The Rebuttal:**

Please address the major concerned in Weaknesses, I am willing to improve the score if the author can provide a detailed explanation on the masking mechanism and discuss how this mechanism will potentially influence the model outcome.

---

> ### Author Response · Authors · 2026-01-25
>
> We are thankful to the reviewer for their insightful feedback. Below, we provide detailed responses to each comment and describe the changes made in the manuscript accordingly.
>
> ---
>
> **The baselines are tasked with simultaneously localizing and describing pathologies, whereas CWCD is given the localization for free.**
>
> 1. To establish a fair comparison, we include an additional experiment (See appendix E.) where we provided the LLaVA-Rad baseline with ground truth visual prompts (VPs) in two ways: (i) alpha blended VPs applied directly to the input X-ray following prior work (Zhu et al., Denner et al.), and (ii) masked visual prompts used in combination with vocabulary subselection for contrastive decoding, effectively extending the approach of Wan et al. As shown in Tab. 6, even with this additional information, their performance (2nd and 3rd row) remains below category-wise report generation (CW, 4th row) and CWCD, demonstrating that the observed improvements are not due to receiving localization information, but rather due to the category-wise parametrization and contrastive decoding framework itself.
>
> 2. We would like to further clarify that masked bounding boxes do not provide pathology localization to CWCD. The masked visual prompts are used to obfuscate information from CWCD, generating a prior biased log probability distribution which is then contrasted with the base log probability distribution generated without any visual prompts or localization information. For example, if an X-ray contains a pacemaker, the masked branch will have the pacemaker region masked out, providing no information about its presence, whereas the base branch receives the standard X-ray without any visual prompts (see Fig. 3). By contrasting these two distributions, CWCD effectively removes prior bias from the base distribution without giving either branch explicit localization of the pacemaker or any other pathology.
>
> 3. **Summary.** In short, CWCD does not receive explicit localization guidance. The masked visual prompts serve solely to facilitate contrastive learning by removing prior bias, and we demonstrate that performance improvements arise from the category-wise report generation approach and contrastive decoding, not from privileged access to pathology locations.
>
> ---
>
> **The details of how the bounding boxes were generated.**
> We have expanded the 2nd paragraph of Sec. 3.1 to include additional details about how the ground truth bounding boxes were generated.
>
> ---
>
> **Explanation of the masking mechanism and how this mechanism will potentially influence model outcome.**
>
> 1. While generating a structured radiology report for a particular category, all pixels on and within the bounding boxes are blacked out (RGB value of 0,0,0), effectively hiding all underlying information in those pixels from the MLLM as seen in Fig. 1. This leads the model to produce tokens conditioned on other areas of the X-ray as well as the prior text tokens.
>
> 2. This masking mechanism is critical for contrastive decoding, as it enables a controlled comparison between tokens produced with and without access to the relevant visual region. By fully removing category-specific visual evidence, differences in the resulting outputs reflect the model’s reliance on that region for generating category-specific descriptions. Partial masking or soft attenuation may allow residual visual cues to persist, weakening the contrastive signal. Therefore, complete masking provides a clear intervention for isolating the contribution of the masked region to the generated text.
>
> We have included the information about the masking mechanism in the appendix F.
>
> ---
>
> **References**
>
> - Kangyu Zhu, Ziyuan Qin, Huahui Yi, Zekun Jiang, Qicheng Lao, Shaoting Zhang, and Kang Li. 2025. *Guiding Medical Vision-Language Models with Diverse Visual Prompts: Framework Design and Comprehensive Exploration of Prompt Variations*. In Proceedings of the 2025 Conference of the Nations of the Americas Chapter of the Association for Computational Linguistics: Human Language Technologies (Volume 1: Long Papers), pages 11726–11739, Albuquerque, New Mexico. Association for Computational Linguistics.
>
> - Denner, Stefan, et al. *Visual Prompt Engineering for Vision Language Models in Radiology.* MIDL 2025
>
> - David Wan, Jaemin Cho, Elias Stengel-Eskin, and Mohit Bansal. *Contrastive region guidance: Improving grounding in vision-language models without training.* In Computer Vision – ECCV 2024, pages 198–215, Cham, 2025. Springer Nature Switzerland.

---

### Author Rebuttal · Authors · 2026-01-25

**Rebuttal:**

We are grateful to the reviewers R1 (YcHF10), R2 (uBUw09), and R3 (S4Tp09) for reviewing our manuscript and highlighting its clear motivation, significant improvements over prior baselines and well defined ablation studies. Their valuable feedback and suggestions have helped us in improving our work.

---

**In summary,**

- We addressed R1’s concern about fairness in our evaluation by adding another experiment in appendix E.
- We evaluated our model on an additional OOD dataset as suggested by R2.
- We addressed R3’s concerns about requiring access to ground truth bounding boxes by fine-tuning Grounding DINO and using bounding boxes generated by it to achieve strong performance on the OOD dataset.
- We clarified R3’s concerns about novelty by highlighting the fundamental problem of hallucination in MLLMs solved through our lightweight category-specific parametrization method as opposed to more complex and training heavy reinforcement learning based prior approaches.

---

We have highlighted all changes in red text in the updated manuscript. Below, we mention some notable changes:

- Evaluation on out-of-distribution (OOD) dataset (Sec. 3.6)
- Addressing fairness concern in our evaluation (Appendix E)
- Masking mechanism (Appendix F)
- Extended Motivation (Appendix A)
- Expanded Dataset description (Appendix C and Sec. 3.1)

**Supporting Material:**

/attachment/abe9398907fce53621ba2c7e0cebfb51fe1081b5.zip

---

### Meta-Review · Area_Chair_PV8n · 2026-02-13

**Recommendation:** Accept (Oral)
**Confidence:** 5

**Metareview:**

Reviewers consistentantly provide postive reviews

---

### Decision · Program_Chairs · 2026-02-13

Accept (Poster)